# ENHANCEMENT-DRIVEN PRETRAINING FOR ROBUST FINGERPRINT REPRESENTATION LEARNING

## ABSTRACT

Fingerprint recognition stands as a pivotal component of biometric technology, with diverse applications from identity verification to advanced search tools. In this paper, we propose a unique method for deriving robust fingerprint representations by leveraging enhancement-based pre-training. Building on the achievements of U-Net-based fingerprint enhancement, our method employs a specialized encoder to derive representations from fingerprint images in a self-supervised manner. We further refine these representations, aiming to enhance the verification capabilities. Our experimental results, tested on publicly available fingerprint datasets, reveal a marked improvement in verification performance against established self-supervised training techniques. Our findings not only highlight the effectiveness of our method but also pave the way for potential advancements. Crucially, our research indicates that it is feasible to extract meaningful fingerprint representations from degraded images without relying on enhanced samples.

## 1 INTRODUCTION

Fingerprint recognition remains one of the most widely used biometric identification methods due to its uniqueness, permanence, and user-friendliness (Maltoni et al., 2022; Wayman et al., 2005; Allen et al., 2005). As applications in law enforcement, personal identification, and secure authentication continue to surge, enhancing the precision and efficiency of fingerprint recognition systems has grown more crucial (Allen et al., 2005).

Although strides have been made in the field, certain challenges persist. These include handling partial or distorted fingerprints caused by the presence of noise or acquisition errors, addressing high interclass similarity, and managing the expansive dimensionality of the feature space (Maltoni et al., 2022; Hong et al., 1998; Cappelli et al., 2007). In the midst of these challenges, many previous works demonstrating state-of-the-art results in the area of fingerprint matching and verification employ minutia-based matching approaches (Ratha et al., 1996; Chang et al., 1997; Maltoni et al., 2022; Cappelli et al., 2010b;a; Jain et al., 2001; 1997). To match two fingerprint images or in other words, to find the similarity between two fingerprint images, the process typically involves extracting minutiae from the images and then these minutiae templates are matched to get a similarity score. If this score surpasses a specific threshold, the two fingerprints are deemed to be identical else they are considered distinct (Maltoni et al., 2009). While minutiae-based methods are traditional in fingerprint recognition, their limitations, such as noise sensitivity, difficulty with partial prints, and susceptibility to non-linear distortions, are pronounced. These issues become evident in scenarios such as poor-quality images leading to the erroneous identification of minutiae (Hong et al., 1998; Maltoni et al., 2009; Zaeri, 2011). In contrast, Convolutional Neural Networks (CNNs) have emerged as effective tools, overcoming these limitations and improving the accuracy and dependability of fingerprint recognition tasks. They efficiently handle partial prints and are tolerant to distortions and various finger conditions by learning local fingerprint structures and adapting to diverse data, including unfavorable finger states like dryness or wetness (Nguyen et al., 2018; Deshpande et al., 2020; Darlow & Rosman, 2017; Tang et al., 2017; Engelsma et al., 2019). Moreover, CNNs streamline the recognition process by integrating feature extraction and matching, enhancing optimization. With the capability to learn fixed-length fingerprint embeddings, CNNs showcase scalability, offering efficient comparison and indexing even as the database size grows (Engelsma et al., 2019). These attributes underscore CNNs as advanced, practical alternatives in fingerprint processing, circumventing the constraints of minutiae-based approaches.

Recently, the field of machine learning has witnessed a surge in the popularity of self-supervised learning techniques (Jaiswal et al., 2020; Liu et al., 2021; Jing & Tian, 2020). In the context of fingerprint biometrics, self-supervised learning promises a solution to the problems inherent to data acquisition for supervised learning. It offers a pathway to learn meaningful representations from abundant unlabeled fingerprint data, bypassing the need for time-consuming and labor-intensive acquisition and labeling processes. Furthermore, self-supervised learning can potentially capture more intricate, data-specific patterns, leading to richer, more robust fingerprint representations.

In this paper, we explore the potential of deep CNNs to learn robust fingerprint representations that can achieve superior matching performance. Building on prior works that introduced U-Net-based fingerprint enhancement methods (Gavas & Namboodiri, 2023; Qian et al., 2019; Liu & Qian, 2020), we propose a pretraining technique centered on fingerprint enhancement. The U-Net model, originally designed for biomedical image segmentation (Ronneberger et al., 2015), has demonstrated efficacy in fingerprint enhancement due to its ability to extract contextual information from input fingerprints and generate enhanced prints that retain intricate structural details. Leveraging the encoder part of U-Net, especially the bottleneck layer, we aim to derive compact and discriminative fingerprint embeddings. We suggest that these refined embeddings could significantly enhance fingerprint search and verification tasks by augmenting class separability and introducing invariance to noise and other distortions.

Our study pursues two main objectives: First, we propose a pretraining technique with U-Net, optimizing a fingerprint enhancement task, and then use this pre-trained encoder to learn fingerprint representations effectively. Second, we plan to assess the efficacy of these representations by comparing their verification performance against existing self-supervised methods. In doing so, we experiment with training and inference techniques to optimize the use of representations for fingerprint verification tasks.

This paper delves into a thorough exploration of these objectives, with the goal of providing insights to deepen our understanding of fingerprint recognition and inspire future progress in this direction.

## 1.1 CONTRIBUTIONS

Here are the main contributions of this work:

1. We suggest a pre-training technique with the U-Net encoder on fingerprint enhancement task and demonstrate the usefulness of this approach in representation learning in a self-supervised setting.
2. We describe a method to fine-tune the learned embeddings for fingerprint verification task.
3. We evaluate our approach with various evaluation metrics demonstrating its effectiveness in fingerprint verification task and also provide a comparison with previous state-of-the-art self-supervised learning methods.

## 1.2 RELATED WORK

The need for better fingerprint recognition tools has led to many developments in learning how to represent fingerprints effectively. Over time, various methods have been introduced, each improving the accuracy and speed of identifying fingerprints in their own way. Many works have combined different strategies focusing on domain knowledge for effective learning of fingerprint representations (Engelsma et al., 2019; Tang et al., 2017). In this paper, we explore enhancement task for pre-training our model for representation learning.

**Image Enhancement**

Early work in fingerprint image enhancement employed techniques such as Gabor filters (Greenberg et al., 2002; Hong et al., 1998; Kim et al., 2002; Yang et al., 2002; Liu et al., 2014), Fourier Transform (Sherlock et al., 1992; Chikkerur et al., 2005; Rahman et al., 2008), and ridge pattern analysis for image enhancement and feature extraction. While these methods seemed promising, they were often challenged by poor image quality, noise, and variations in ridge patterns.

Deep learning techniques, particularly Convolutional Neural Networks (CNNs), have increasingly been used in fingerprint recognition. CNNs, due to their ability to learn hierarchical representa-

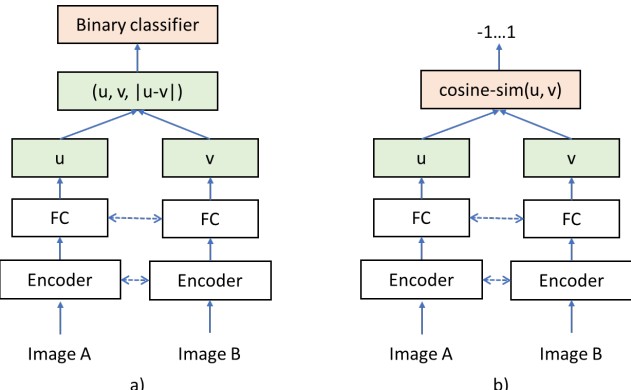

Figure 1: a) Architecture with verification objective i.e with binary classifier (at training and inference) b) Architecture to compute similarity scores (at inference). The dotted arrows indicate networks having tied weights (siamese network structure).

tions, have been effective in capturing minutiae and latent features in fingerprints, thereby improving recognition accuracy (Nguyen et al., 2018; Deshpande et al., 2020; Tang et al., 2017).

Originally developed for biomedical image segmentation (Ronneberger et al., 2015), U-Net is an encoder-decoder network that excels in tasks that require contextual information from the input image. A significant development in the fingerprint enhancement domain was the adaptation of the U-Net architecture for fingerprint enhancement. Gavas & Namboodiri (2023) proposed multiple modifications to U-Net architecture tailored for the fingerprint domain for the enhancement task incorporating minutia and orientation knowledge in the network. In their work, Qian et al. (2019) introduced a fingerprint enhancement approach using a deep network called DenseUNET to improve image quality in a pixel-to-pixel and end-to-end manner. Liu & Qian (2020) used U-Net for segmentation and enhancement of latent fingerprints.

**Self-supervised Learning Techniques**

Self-supervised learning has emerged as a compelling alternative to traditional supervised learning, particularly in contexts marked by scarcity or complexity of labeled data (Jaiswal et al., 2020; Jing & Tian, 2020). This learning paradigm capitalizes on the vast availability of unlabeled data, using cleverly designed pretext tasks to derive useful feature representations. Generally, these techniques learn by predicting certain aspects of the input data based on other parts, or by solving auxiliary tasks.

Contrastive learning, based on the principle of learning by comparison, stands as a cornerstone of self-supervised learning (Liu et al., 2021). It revolves around the idea of differentiating between similar (positive) and dissimilar (negative) instances. Various techniques that illustrate this concept have achieved significant success. The SimCLR (Chen et al., 2020a) framework, for instance, produces augmented versions of an image, treats them as distinct instances, and then trains the model to identify the original image pair among a set of negative samples. Another example is MoCo (Momentum Contrast) (Chen et al., 2020b) which maintains a dynamic dictionary of data samples in a queue and a moving-averaged encoder to tackle the challenge of large-scale instance discrimination. More recent advancements include methods like BYOL (Bootstrap Your Own Latent) (Grill et al., 2020) which deviate from the traditional contrastive learning paradigm by not using negative samples and instead focusing on bringing representations of different views of the same image closer in the embedding space. Similarly, SwAV (Caron et al., 2021) uses a unique approach of swapping cluster assignments to maximize consistency between differently augmented views of the same image. Another notable technique is the Noise Contrastive Estimation (NCE) (Gutmann & Hyvärinen, 2010), which contrasts a true data sample against noise samples and has been extensively used in the field of natural language processing for word embedding learning. These techniques provide a comprehensive insight into contrastive learning, a critical approach within self-supervised learning that has potential applications in fingerprint biometrics.

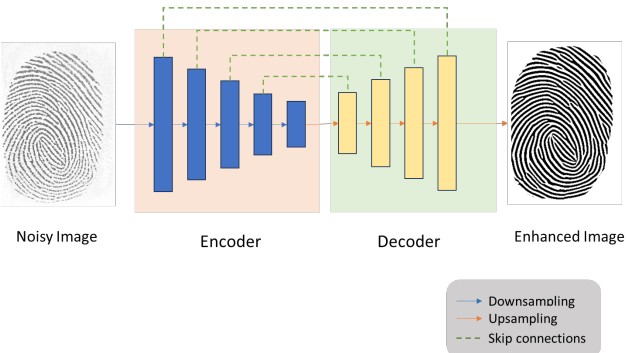

Noisy Image  Encoder  Decoder  Enhanced Image

→ Downsampling
→ Upsampling
--- Skip connections

Figure 2: U-Net architecture for enhancement task for the pre-training stage in the self-supervised setting. For representation learning, the decoder is discarded and the binary classifier is attached.

## 2 METHODOLOGY

The methodology for our research is constructed around a two-stage framework to probe the potential of self-supervised learning in fingerprint representation learning. A broad overview of the process is as follows:

- **Stage 1: Self-Supervised Pre-training:** This is the initial stage of our methodology, in which we perform pre-training of our models in a self-supervised manner. It includes the application of both existing self-supervised learning techniques as well as our novel enhancement-based approach for this task. The intent of this stage is to leverage the power of unlabeled data to learn meaningful representations that can serve as a starting point for subsequent stages. Notably, for all methods, we keep the encoder architecture the same. While other self-supervised methods traditionally use encoders like ResNet or Vision Transformers, in our framework we use the encoder of our U-Net-based model to ensure a fair comparison.

- **Stage 2: Probing Experiments:** Upon completion of the pre-training phase, we progress to the second stage where a few linear layers (MLP) are added on top of the frozen pre-trained encoder, making the representations 512-d. We then perform probing experiments using this newly formed model. By keeping the encoder part frozen, we ensure that the model adapts the existing representations for the verification task without altering the learned patterns from the self-supervised pre-training phase.

Following this framework, we navigate through the process of adapting and implementing self-supervised learning techniques, exploring a U-Net-based pre-training strategy, and conducting probing experiments with pre-trained networks. The sections below provide a detailed overview of the procedures involved in each stage.

### 2.1 U-NET-BASED PRETRAINING

While the adaptation and application of existing self-supervised methods to fingerprint data provide a valuable starting point, we believe that the uniqueness of fingerprint data could benefit from a self-supervised learning method specifically tailored for it. Building upon our findings in U-Net-based enhancement works, our approach leverages the training of a fingerprint enhancement model as a form of self-supervision.

We explore the use of a U-Net-based fingerprint enhancement as a pre-training strategy. We hypothesize that the U-Net encoder, trained on the task of fingerprint enhancement, might contain valuable fingerprint representations. The process of enhancing a fingerprint image can serve as an effective self-supervised task, encouraging the model to learn useful, fingerprint-specific representations. Specifically, the pre-trained encoder already encapsulates valuable information about the fingerprint, which can be used as a foundation for further representation learning. It's crucial to note that the quality of these initial representations heavily relies on the effectiveness of the U-Net-

based enhancement model. Hence, the importance of the enhancement model's design and training discussed here, cannot be overstated. For the enhancement-based pre-training stage, we utilize the basic U-Net architecture in Figure 2 to optimize the fingerprint enhancement task. This is a simple image-to-image network where the input of the network is a fingerprint image which is generally degraded with various kinds of noises. The network tries to learn to predict an enhanced version of the fingerprint image by removing noise as much as possible such that the ridge structure of the fingerprint is maintained and restored. This ensures that the network learns the minute details of fingerprint structure and also tries to enhance it wherever possible which proves helpful in extracting robust feature representations in later stages.

## 2.2 LEARNING FINGERPRINT REPRESENTATION

After the self-supervised pre-training, we conduct the probing experiments using the pre-trained networks. The aim of these experiments is to assess the usefulness of the learned representations for the task of fingerprint verification. For this, we add 3-layer MLP projection head on top of the frozen encoder part of the pre-trained network. We then train this model using a Sentence-BERT-like (Reimers & Gurevych, 2019) siamese architecture, with a limited amount of labeled data for the fingerprint verification task. We concatenate the fingerprint representations $u$ and $v$ of the image pair with the element-wise difference $|u - v|$ and then pass it through the linear layers and train it for binary-classification objective as illustrated in Figure 1. By keeping the encoder part frozen, the model learns to adapt the existing representations for the verification task, without changing the underlying learned patterns. This approach allows us to leverage a large amount of unlabeled data to learn initial representations and a limited amount of labeled data for supervised adaptation. Note that in the supervised fine-tuning, allowing modifications in the encoder weights can lead to higher performance on the end task, which is the future scope of this work. As our goal here is to examine the robustness of the learned representations by different pre-training techniques, we keep the encoder frozen. In summary, the combination of self-supervised pre-training with supervised fine-tuning offers a promising learning framework for fingerprint biometrics. Our methodology aims to leverage the strengths of both self-supervised and supervised learning, offering a pathway towards robust, efficient, and data-savvy fingerprint biometrics systems.

## 3 EXPERIMENTS

In this section, we discuss the experiments performed to evaluate our proposed approach's efficacy. We cover the specifics of our experimental setup, including the datasets used, the training details, and the evaluation metrics employed.

## 3.1 DATASETS AND PREPROCESSING

The datasets used in this study consist of both synthetic and real-world fingerprint images, originating from the Synthetic Fingerprint Generator (SFinGe) (Cappelli, 2004), the Fingerprint Verification Competition (FVC) (Maio et al., 2002b;a; 2004), and the NIST SD-302 database (Fiumara et al., 2019).

Table 1: Summary of Datasets

| Dataset | Source | Identities | Images | Purpose |
|---------|--------|-----------|--------|---------|
| SFinGe | Synthetic | 3,700 | 15000 | Train |
| FVC-2000 | Real-world | 440 | 3520 | Train |
| FVC-2002 | Real-world | 440 | 3520 | Train |
| NIST SD-302 | Real-world | 2000 | 8000 | Train |
| SFinGe | Synthetic | 1584 | 6336 | Test |
| FVC-2004 | Real-world | 440 | 3520 | Test |

The synthetic SFinGe dataset was crafted to reflect real-world challenges in fingerprint recognition, such as various backgrounds, rotation degrees, and the presence of scratches and noise. The real-world FVC datasets and NIST SD-302 database provide large-scale, realistic fingerprint data,

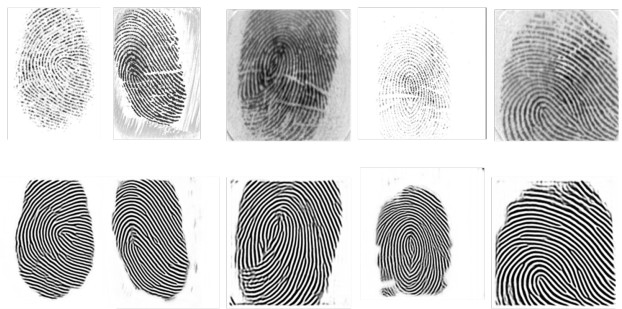

Figure 3: Degraded-Enhanced pairs on FVC dataset from enhancement pre-training

ensuring the generalizability and robustness of the findings. This combination of synthetic and real-world datasets allows the training and evaluation of the model under a diverse set of conditions. The synthetic data offers scalability and control over experimental parameters, while the real-world data ensures applicability in realistic conditions. Table 1 provides summary statistics of the datasets.

For the self-supervised pre-training phase, we use only the fingerprint images from the training datasets, disregarding any associated ground truth labels or identities, in accordance with the principles of self-supervised learning. But we still require ground-truth for the enhancement task. For the SFinGe dataset, the corresponding clean images are already available for the degraded fingerprints which we use directly. For NIST SD 302 and FVC datasets, we generate enhanced fingerprints from degraded fingerprints with a classical approach as described in work by Hong et al. (1998). In the next stage of our approach, the training process shifts to a binary classification task for fingerprint verification. Positive and negative fingerprint pairs are generated taking into account their associated identities. In this stage, we skip the NIST SD 302 as the impressions for fingerprints vary a lot due to sensor variations and acquisition conditions which worsens the training. As data augmentation is crucial for a self-supervised paradigm, we apply a random combination of transformations like rotation, color jitter, resize, crop, and Gaussian blur for the data augmentation.

## 3.2 Implementation Details

We conduct all our experiments using the PyTorch (Paszke et al., 2017) deep learning framework with Nvidia GeForce RTX 2080 Ti GPU for training.

Our proposed enhancement-based pre-training technique utilizes the U-Net architecture as discussed in the previous section. The same U-Net encoder architecture is used for pre-training with other self-supervised learning methods for a fair comparison with our technique. During self-supervised pre-training, all the models are trained from scratch. Our U-Net architecture has a depth of 5 layers, wherein each layer consists of 2 convolutions. It expects gray-scale fingerprint images with dimensions of 512 x 512 pixels as encoder input and decoder output. Therefore, all input images were resized and padded as required to match this input size. The encoder produces a 4096-dimensional vector bottleneck which is reduced to 512-d with MLP projection head. In addition, the model uses depth-wise convolutions to reduce the overall number of parameters. For our approach, we use the $L_2$ or MSE (Mean Squared Error) loss for the enhancement-based pre-training of U-Net model. For other self-supervised techniques, the losses are the same as described in their papers.

For the pre-training with the existing self-supervised methods, we performed a grid search to identify the best hyperparameters, such as learning rate, batch size, learning rate schedule, and momentum value, and pre-trained the models for 50 epochs with early stopping applied to prevent overfitting.

For probing experiments, we adapt the MLP projection head weights for the verification task while keeping the encoder weights fixed. We generate training and testing verification sets from each dataset in a 1:3 ratio of positive and negative pairs. After training, we evaluate the models on the test sets for the verification task. We report various metrics like verification accuracy, precision, recall, and F1-score to evaluate the performance of different techniques. We report the results in two ways: 1) using the binary classifier over the MLP projection head (similar to the training setup). Refer to Figure 1-a and 2) only utilizing the representations and applying thresholds on the cosine similarity

Refer to Figure 1-b. The formulation of cosine similarity is mentioned in Appendix A.1. The first method simply evaluates the trained model as an end-to-end verification network. The second one shows us the potential to use the learned representations in the context of similarity search and in turn for recognition tasks.

## 3.3 RESULTS

The models are first pre-trained to learn fingerprint representations using the enhancement-based approach and various self-supervised learning strategies. Because these representations are not explicitly trained for fingerprint verification or identification, using them directly for evaluation is inappropriate. To gauge the stability and usefulness of these learned representations, we add linear layers to the frozen pre-trained encoders and then train the models for fingerprint verification tasks. The encoders remain frozen, allowing only the weights of the MLP to adjust to the task, keeping the original representations unchanged. This setup aids in comparing the efficacy of different self-supervised learning techniques against our method. The results of our probing experiments are presented in Table 4 (Verification Accuracy), 5 (Precision) 6 (Recall) and 7 (F1-score). The verification accuracy and F1-score on the SFinGe test set are shown in Tables 2 and 3 respectively. Precision and Recall results are shown in Appendix A.2. Figure 3 shows a few sample pairs of input and predicted images from the pre-trained U-Net model on the enhancement task used in our approach.

Our approach is compared with methods like SimCLR v2, SimSiam, MoCo v2, and BYOL on the SFinGe and FVC test sets for fingerprint verification. Verification accuracy serves as the evaluation metric for each method. The test data for fingerprint verification consists of a 1:3 ratio of positive to negative pairs, setting the random guess accuracy at 75%. Verification accuracy is measured in two ways as described before. This is presented in the below tables under the 'Classifier' column. The second way is represented under the 'Similarity' column in the tables. Moreover, we also report the ROC curves in Figure 4 for both datasets.

As seen from the results, our enhancement-based pre-training method consistently outperforms other self-supervised strategies across both test datasets. SimCLR v2 also consistently performs well. SimSiam and BYOL methods show comparatively poor performance. It is noteworthy that all models perform better on the SFinGe test set than on the FVC test set. We believe this is due to two primary factors: the training sets contain more data from SFinGe than FVC, potentially resulting in a bias towards the former, and SFinGe is a synthetic dataset while FVC consists of real fingerprints, making the latter more challenging. Hence, the performance of models on FVC datasets is the real measure of the efficacy of models. Importantly, our method also provides superior performance when verification is based on the similarity of the representations, suggesting that the learned representations are also useful for fingerprint recognition.

Table 2: Verification accuracy on SFinGe test dataset with genuine and imposter pairs

| SFinGe - Accuracy | | | | | | |
|---|---|---|---|---|---|---|
| **Method** | **Classification** | | | **Similarity** | | |
| | **Imposter** | **Genuine** | **Entire Data** | **Imposter** | **Genuine** | **Entire Data** |
| **SimCLR** | 0.968 | 0.881 | 0.96 | 0.982 | 0.749 | 0.961 |
| **SimSiam** | 0.972 | 0.362 | 0.916 | 0.888 | 0.648 | 0.866 |
| **MoCo** | 0.963 | 0.881 | 0.956 | 0.955 | 0.845 | 0.945 |
| **BYOL** | 0.96 | 0.825 | 0.947 | 0.963 | 0.718 | 0.941 |
| **Ours** | **0.982** | **0.886** | **0.973** | **0.975** | **0.847** | **0.963** |

## 4 LIMITATIONS AND FUTURE WORK

Despite our promising findings, there exist certain limitations that need to be acknowledged. Primarily, the current model is more effective on the SFinGe dataset, which is synthetic, than on the FVC dataset that encompasses real-world fingerprints. This discrepancy might be due to the under-representation of FVC data in the training sets, leading to potential bias, and also due to inherent complexities in real-life fingerprint data, which are harder to handle. Another limitation is that our

Table 3: F1 score on SFinGe test dataset with genuine and imposter pairs

| SFinGe - F1 score | | | | | | |
|---|---|---|---|---|---|---|
| Method | Classification | | | Similarity | | |
| | Imposter | Genuine | Entire Data | Imposter | Genuine | Entire Data |
| SimCLR | 0.98 | 0.8 | 0.803 | 0.98 | 0.78 | 0.781 |
| SimSiam | 0.96 | 0.44 | 0.442 | 0.92 | 0.47 | 0.469 |
| MoCo | 0.98 | 0.79 | 0.785 | 0.97 | 0.74 | 0.737 |
| BYOL | 0.97 | 0.74 | 0.742 | 0.97 | 0.69 | 0.689 |
| Ours | **0.99** | **0.86** | **0.858** | **0.98** | **0.81** | **0.821** |

Table 4: Verification accuracy on FVC test dataset with genuine and imposter pairs

| FVC - Accuracy | | | | | | |
|---|---|---|---|---|---|---|
| Method | Classification | | | Similarity | | |
| | Imposter | Genuine | Entire Data | Imposter | Genuine | Entire Data |
| SimCLR | 0.915 | 0.619 | 0.888 | 0.943 | 0.537 | 0.906 |
| SimSiam | 0.956 | 0.122 | 0.88 | 0.387 | 0.733 | 0.419 |
| MoCo | 0.902 | 0.522 | 0.867 | 0.896 | 0.56 | 0.865 |
| BYOL | 0.886 | 0.568 | 0.857 | 0.926 | 0.477 | 0.886 |
| Ours | **0.957** | **0.73** | **0.937** | **0.933** | **0.818** | **0.923** |

study did not specifically train or evaluate the fingerprint recognition task. While our model has shown potential for this task, a specific and thorough evaluation is needed to fully understand its performance in this regard. Furthermore, the efficacy of self-supervised learning methodologies is inherently reliant on the quality and diversity of the training data. In addition, we have utilized a specific form of linear probing for our study. There exist alternative approaches, such as softmax or ArcFace-based classification, which may help in learning better representations.

In terms of future work, it would be beneficial to address these limitations by including a broader, more diverse range of real-world fingerprint datasets in the training phase. Moreover, instead of freezing the encoder, training it with a smaller learning rate can be explored. This may enhance the model's generalizability and make it more robust against a variety of fingerprint data. Specific training and evaluation for the recognition task, as well as exploring alternative linear probing techniques, would also be valuable directions to pursue. Additionally, other forms of self-supervised learning methods could be explored to continually optimize the model performance.

## 5    CONCLUSION

In this paper, we explored the application of various self-supervised learning techniques for pre-training a model to learn good fingerprint representations which can be useful to recognize and verify fingerprints. We proposed a novel approach to use fingerprint enhancement as a self-supervised pre-training method. We performed probing experiments that proved beneficial in evaluating the effectiveness of learned fingerprint representations across different pre-training strategies. The verification performance of our method was compared against SimCLR v2, SimSiam, MoCo v2, and

Table 5: Precision on FVC test dataset with genuine and imposter pairs

| FVC - Precision | | | | | | |
|---|---|---|---|---|---|---|
| Method | Classification | | | Similarity | | |
| | Imposter | Genuine | Entire Data | Imposter | Genuine | Entire Data |
| SimCLR | 0.96 | 0.42 | 0.422 | 0.95 | 0.49 | 0.486 |
| SimSiam | 0.92 | 0.22 | 0.218 | 0.94 | 0.11 | 0.107 |
| MoCo | 0.95 | 0.35 | 0.347 | 0.95 | 0.35 | 0.35 |
| BYOL | 0.95 | 0.33 | 0.334 | 0.93 | 0.48 | 0.394 |
| Ours | **0.97** | **0.63** | **0.634** | **0.98** | **0.55** | **0.553** |

Table 6: Recall on FVC test dataset with genuine and imposter pairs

| Method | Classification | | | Similarity | | |
|---|---|---|---|---|---|---|
| | Imposter | Genuine | Entire Data | Imposter | Genuine | Entire Data |
| SimCLR | 0.92 | 0.62 | 0.619 | 0.94 | 0.54 | 0.537 |
| SimSiam | 0.96 | 0.12 | 0.122 | 0.39 | 0.73 | 0.733 |
| MoCo | 0.9 | 0.52 | 0.522 | 0.9 | 0.56 | 0.56 |
| BYOL | 0.89 | 0.57 | 0.568 | 0.93 | 0.48 | 0.477 |
| **Ours** | **0.96** | **0.73** | **0.73** | **0.93** | **0.82** | **0.818** |

Table 7: F1 score on FVC test dataset with genuine and imposter pairs

| Method | Classification | | | Similarity | | |
|---|---|---|---|---|---|---|
| | Imposter | Genuine | Entire Data | Imposter | Genuine | Entire Data |
| SimCLR | 0.94 | 0.5 | 0.502 | 0.95 | 0.51 | 0.51 |
| SimSiam | 0.94 | 0.16 | 0.156 | 0.55 | 0.19 | 0.186 |
| MoCo | 0.93 | 0.42 | 0.417 | 0.92 | 0.43 | 0.431 |
| BYOL | 0.92 | 0.42 | 0.421 | 0.94 | 0.43 | 0.432 |
| **Ours** | **0.97** | **0.68** | **0.679** | **0.96** | **0.66** | **0.659** |

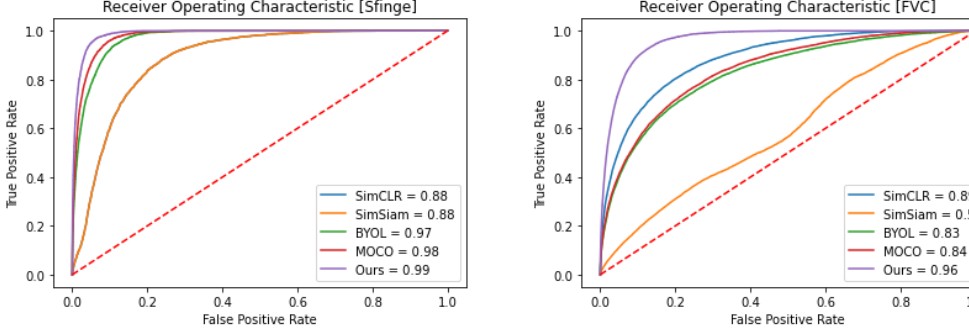

Figure 4: ROC curve based on similarity scores on SFinGe dataset(left) and FVC dataset (right)

BYOL methods using two test sets, SFinGe and FVC. Our method consistently outperformed other techniques across both test datasets, thereby demonstrating the robustness and effectiveness of our model. Our method also surpassed other methods when evaluated in terms of similarity-based verification indicating the effectiveness of the learned representations for fingerprint recognition task. However, it was observed that all models performed better on the synthetic SFinGe dataset compared to the real-world FVC dataset, indicating potential limitations related to bias in the training set and complexities of real-world fingerprint data. In the future, we intend to extend our research to encompass more diverse and complex real-world fingerprint datasets, thus enhancing the generalizability of our model. We also plan to investigate other self-supervised learning methods and strategies to enhance the model's performance and adaptability to real-world fingerprint data complexities. In conclusion, this chapter illuminates the potential of self-supervised learning methods in the domain of fingerprint biometrics, while highlighting areas for further exploration and improvement.

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

## A  APPENDIX

### A.1  COSINE SIMILARITY

$$\text{cosine similarity } (x_1, x_2) = \frac{x_1 \cdot x_2}{\max(||x_1||_2 \cdot ||x_2||_2, \epsilon)} \tag{1}$$

where $x_1$ and $x_2$ are representation vectors and $\epsilon$ is very small value $1e^{-8}$ to avoid division by zero.

### A.2  RESULTS ON SFINGE DATASET

Table 8: Precision on SFinGe test dataset with genuine and imposter pairs

| Method | SFinGe - Precision | | | | | |
|--------|----------|---------|-------------|----------|---------|-------------|
| | Classification | | | Similarity | | |
| | Imposter | Genuine | Entire Data | Imposter | Genuine | Entire Data |
| SimCLR | 0.99 | 0.74 | 0.737 | 0.98 | 0.82 | 0.815 |
| SimSiam | 0.94 | 0.57 | 0.567 | 0.96 | 0.37 | 0.367 |
| MoCo | 0.99 | 0.71 | 0.708 | 0.98 | 0.65 | 0.654 |
| BYOL | 0.98 | 0.67 | 0.674 | 0.97 | 0.66 | 0.662 |
| **Ours** | **0.99** | **0.83** | **0.832** | **0.98** | **0.78** | **0.776** |

Table 9: Recall on SFinGe test dataset with genuine and imposter pairs

| SFinGe - Recall | | | | | | |
|---|---|---|---|---|---|---|
| Method | Classification | | | Similarity | | |
| | Imposter | Genuine | Entire Data | Imposter | Genuine | Entire Data |
| SimCLR | 0.97 | 0.88 | 0.881 | 0.98 | 0.75 | 0.749 |
| SimSiam | 0.97 | 0.36 | 0.362 | 0.89 | 0.65 | 0.648 |
| MoCo | 0.96 | 0.88 | 0.881 | 0.96 | 0.85 | 0.845 |
| BYOL | 0.96 | 0.83 | 0.825 | 0.96 | 0.72 | 0.718 |
| Ours | **0.98** | **0.89** | **0.886** | **0.98** | **0.85** | **0.847** |

