# OpenReview forum: "Enhancement-Driven Pretraining for Robust Fingerprint Representation Learning"
_ICLR.cc/2024/Conference — ICLR 2024 Conference Withdrawn Submission_

### Official Review · Reviewer_eThS · 2023-10-27

**Soundness:** 3 good
**Presentation:** 3 good
**Contribution:** 2 fair
**Rating:** 3
**Confidence:** 3

**Summary:**

This paper deals proposes a  fingerprint enhancement approach and a method to obtain fingerprint image representations in a self-supervised manner. The proposed method is trained using a combination of synthetic fingerprints as well as real fingerprints and tested on both real and synthetic fingerprints. The accuracy metrics indicate that the proposed approach outperforms other comparable approaches and that the verification performance is better on the synthetic fingerprints compared to the real fingerprints.

**Strengths:**

The main strength of the paper is its set of experimental results. They indicate that the proposed approach provides better accuracies than other comparable approaches, such as SimCLR, SimSiam, MoCo and BYOL.

Another strength of the paper is that it is easy to understand and the quality of the language is good.

**Weaknesses:**

This manuscript has following high-level weaknesses that need to be addressed in any revision.

1. Fingerprint verification accuracy on FVC test dataset is worse than on SFinGe test dataset. While I agree with the authors' explanation that it "might be due to underrepresentation of FVC data in the training sets, leading to potential bias, and also due to inherent complexities in real-life fingerprint data, which are harder to handle", these challenges are faced by all learning-based fingerprint recognition approaches and solutions that don't perform well on real data will be less attractive.

2. Perhaps this is a weakness for deep learning based fingerprint recognition systems, but given that there are many real-world fingerprint verification systems based on minutiae matching, there needs to be a stronger case made for how the proposed deep learning approach is expected to have an impact in the real world.

3. It is not sufficiently explained what "self supervision" means in this paper. My understanding is that both degraded and enhanced fingerprints are used for training. Aren't the enhanced fingerprints akin to having ground truth and if so isn't that training supervised? Not enough details are provided to understand this correctly.

4. The paper needs to provide more mathematical details of the proposed approach. For example, more details should be provided about the fingerprint enhancement procedure and results.

5. It is not clear what attributes of fingerprints are being taken advantage of, in the proposed approach. Are fingerprint geometric structures or minutiae being used somehow in the training? Overall, this paper is lacking in technical details.

**Questions:**

1. Please be more clear about what self-supervision refers to.

2. Provide more details about the fingerprint enhancement process. How is the enhancement quality evaluated?

3. In Fig. 3, I assume that degraded images are on top and enhanced ones on the bottom  --- if this is correct, please state that in that figure caption.  It appears that the fingerprint degradation is artificial and not based on real-world challenges. If that is correct, could it be the reason for the poorer performance on FVC datasets?

4. I am a little confused about how the accuracy numbers on the entire dataset are computed. In Table 2, it is stated that the accuracy of the proposed method on the entire data is 0.973. But if the impostor accuracy is 0.982 and the genuine accuracy is 0.886 and the data split is 1 genuine pair for every 3 impostor pairs, shouldn't the overall accuracy be (3*0.982 + 1*0.886)/4 = 0.958, not 0.973?

**Details Of Ethics Concerns:**

No concerns

---

> ### Author Response · Authors · 2023-11-22
> **Thank you for the review comments and the efforts**
>
> Thanks to the reviewer for the review comments and appreciate the efforts. We would like to address the reviewer's questions.
> - As we are using enhancement task for pre-training stage with larger dataset of degraded-enhanced image pairs and no identity labels, we are using indirect supervision for verification task. Hence, we are using less labeled samples for verification. We refer to self-supervision in this sense.
> - Enhancement task is denoising process where input is degraded image to U-Net and output is enhanced image. We evaluated the model using SSIM, RSME, PSNR metrics to check image quality. Moreover, we also checked NFIQ2 scores for few images (NFIQ2 is NIST tool that gives score to measure fingerprint image quality). We will update these results and add missing technical details in our paper. Thank you for pointing it out.
> - Yes, first row are degraded images and second row are enhanced counterparts. Sure, we would mention the same in the caption. The images in FVC dataset as the official website says, are collected from sensors by volunteers, hence artificial degradations are not added.
> - We apologize as there is a typo as it is supposed to be 1:10 genuine-imposter pairs. We will correct this in the paper. Thanks to the reviewer for highlighting this.

---

### Official Review · Reviewer_NFpv · 2023-10-29

**Soundness:** 1 poor
**Presentation:** 3 good
**Contribution:** 1 poor
**Rating:** 1
**Confidence:** 4

**Summary:**

In this paper, the authors proposed a self-supervised learning scheme based on image enhancement for learning fingerprint representation. Several standard SSL schemes are compared with the proposed methods. The experimental results show that the proposed method outperforms the standard SSL schemes.

**Strengths:**

The proposed method is simple and easy to implement.
The presentation is clear.

**Weaknesses:**

Collecting human fingerprints require IRB approval. In general, when collecting fingerprint, a unique userID, will be assigned to each subject. Thus, from the point of data collecting, each fingerprint image is given an userID. It is not clear why we need to apply SSL.

Many companies and governments have very large fingerprint databases with the corresponding user IDs. Thus, they can be used to develop highly accurate fingerprint recognition systems based on supervised learning and traditional pattern recognition methods.

Although the experimental results show that the proposed method is better than other SSL methods, the accuracy is still very low comparing with other state-of-the-art fingerprint recognition systems.

In fact, fingerprint recognition for commercial applications is generally considered as a solved problem. The accuracy of the current fingerprint recognition systems is very high.

The novelty of the work is low, comparing with other ICRL paper I read and review. In addition, the topic is more suitable for biometric journals or conferences such as T-Bio, IJCB etc.

**Questions:**

N.A.

---

> ### Author Response · Authors · 2023-11-22
> **Thank you for the review comments and the efforts**
>
> We would like to thank the reviewer for the review comments and efforts. Here, we would like to address reviewer's few concerns.
> - The availability of genuine-imposter pairs (based on number of impressions available for same fingerprint) is generally less in real world datasets. Hence, we use semi-supervised paradigm in a way that we are not directly using supervision from the verification task but we are using enhancement task for pre-training (training dataset is large here). We later use linear probing to finetune on smaller verification task (by using identity labels)
> - We understand the fingerprint recognition accuracy for SOTA methods is high. Our work provides a direction to explore self-supervised enhancement pre-training approach to fingerprint representation learning, where available literature is almost negligible.

---

### Official Review · Reviewer_dTWo · 2023-10-30

**Soundness:** 2 fair
**Presentation:** 3 good
**Contribution:** 2 fair
**Rating:** 3
**Confidence:** 4

**Summary:**

The paper introduces a novel self-supervised learning approach for fingerprint recognition using fingerprint enhancement. Through experiments, the proposed method consistently outperforms established techniques on the SFinGe and FVC datasets. However, performance varies between the synthetic and real-world datasets. The authors acknowledge these limitations and suggest future research directions to enhance model robustness and generalizability.

Key Contributions:
A new self-supervised pre-training method using fingerprint enhancement.
Evaluation against established techniques on two datasets.
Identification of model limitations and suggestions for future research.

**Strengths:**

Originality: The paper presents a novel approach to fingerprint enhancement using self-supervised pre-training methods. The idea of leveraging fingerprint enhancement as a self-supervised pre-training method is innovative and has not been extensively explored in the literature.

Quality: The paper provides a comprehensive set of experiments, comparing the proposed method with several state-of-the-art techniques. The inclusion of both synthetic (SFinGe) and real-world (FVC) datasets in the evaluation provides a holistic view of the model's performance.

Clarity: The paper is well-structured and written. The methodology section provides a clear explanation of the proposed approach, and the results are presented in a coherent manner with tables and figures that aid in understanding the findings.

Significance: The potential application of self-supervised learning techniques in the domain of fingerprint biometrics is significant. The paper's findings could pave the way for more robust and efficient fingerprint recognition systems in the future. The consistent outperformance of the proposed method over other techniques on both test datasets underscores its potential impact in the field.

**Weaknesses:**

Dataset Discrepancy: The paper's models show a significant performance difference between the synthetic SFinGe dataset and the real-world FVC dataset. This discrepancy suggests that the model might not generalize well to real-world data. It's crucial for the model to perform consistently across various datasets, especially in biometric applications where real-world data is the primary concern.

Limited Dataset Representation: The potential bias due to the underrepresentation of the FVC data in the training sets is a significant concern. Relying heavily on one dataset can lead to overfitting and reduced model robustness. Expanding the dataset variety would likely improve the model's generalizability.

Evaluation Scope: The study did not specifically train or evaluate the fingerprint recognition task, which seems to be a core application of the proposed method. A more targeted evaluation on this task would provide clearer insights into the model's practical utility.

Linear Probing Technique: The authors used a specific form of linear probing for their study. While they acknowledge the existence of alternative approaches, such as softmax or ArcFace-based classification, these were not explored. These alternative methods might offer better representation learning, and their omission is a missed opportunity.

Comparative Analysis: While the paper compares the proposed method against established techniques, it would be beneficial to see a deeper analysis of why their method outperforms others. Understanding the underlying reasons for performance differences can guide future improvements.

Self-supervised Learning Reliance: The efficacy of the proposed self-supervised learning methodologies is heavily reliant on the quality and diversity of the training data. This dependence can be a limitation if diverse and high-quality data is not available.

**Questions:**

Dataset Selection:
Why was there a heavier reliance on the synthetic SFinGe dataset over the real-world FVC dataset during the training phase? How does this choice impact the model's performance on real-world data?

Model Generalization:
How do you anticipate the model will perform on other real-world fingerprint datasets not covered in this study? Are there plans to test on additional datasets?

Evaluation Scope:
Why wasn't the fingerprint recognition task specifically trained or evaluated, given its apparent importance to the study's objectives?

Linear Probing Technique:
What led to the decision to use a specific form of linear probing? How do you think alternative techniques might impact the results?

Model Comparison:
While the proposed method outperforms other techniques, can you provide more insights into the specific features or components of your model that contribute to this superior performance?

Self-supervised Learning:
Given the heavy reliance on self-supervised learning, how do you envision the model's performance if trained with a more diverse set of self-supervised learning techniques?

Future Work:
You mentioned potential future directions, including not freezing the encoder and training it with a smaller learning rate. Have preliminary tests been conducted in this direction, and if so, what were the findings?

Model Robustness:
How does the proposed model handle noisy or low-quality fingerprint data, especially given the challenges with real-world data?

Implementation Details:
Could you provide more details about the hyperparameters used during training and any specific architectural choices made for the model?

---

> ### Author Response · Authors · 2023-11-22
> **Thank you for the review comments and the efforts**
>
> Thank you very much for providing detailed review for our work. We would like to address your questions.
> - Dataset selection: Many of the earlier available fingerprint datasets are revoked from public domain. Moreover, they are small and contain very less number of impressions per fingerprint, making us rely on synthetically generated fingerprints. Also, they don't possess enhanced/clean impressions for ground-truths in enhancement pre-training stage.
> - Model Generalization: The models are tested on two standard publicly available test sets NIST SD 302 and FVC datasets used in most of the literature works.
> - Evaluation Scope: We hypothesized that the improved performance on verification task would result in improved performance for recognition task because matching is the crux between both the tasks. Verification task is 1:1 matching whereas Recognition is 1:N matching. Hence, we haven't shown recognition results.
> - Linear Probing Technique: We wanted to test the performance of the learnt representations in comparision with other SOTA self-supervised techniques which normally use projection head on top of encoder. Hence, we employed similar technique for our enhancement-based approach.
> - Model Comparison: As the model was pretrained on enhancement task, it learned the specific details of fingerprints like ridge/valley structures with noise removal, while other methods performance heavily depend on data augmentations and were mostly designed to work for real world natural images.
> - Self-supervised Learning: We employed simpler and intuitive technique in self-supervised paradigm for our approach and yet see considerable performance improvement. Hence, we believe more advanced techniques can improve the results further.
> - Future Work: We haven't performed preliminary experiments as such but we think that keeping encoder unfrozen with very small learning rate can help model perform well on verification dataset which would make sense only when dataset is big enough for model generalizability and avoid overfitting (we possess smaller labeled dataset for second stage training (verification task)).
> - Model Robustness: We evaluated our model on real world fingerprint datasets FVC which contain degraded and noisy fingerprint data. We think that the performance trend should be similar on other real world datasets too, in any.
> - Implementation Details: All algorithm specific hyperparameters for other self-supervised techniques were according to the algorithm papers. LR was set as 1e-3 with batch size of 64 for our approach and U-Net architecture was used as base network.